Freshwater trematodes differ from marine trematodes in patterns connected with division of labor

Neal Allison T. aneal1@norwich.edu
Stettner Moira
Ortega-Cotto Renytzabelle
http://orcid.org/0009-0000-7777-9540 Dieringer Daniel
Reed Lydia C.
Norwich University , Northfield, VT , United States
Kamel Bishoy
Electronic publication date: 2024 Apr 12
Publication date: 2024
Volume: 12
Electronic Location ID: e17211
Received 2023 Oct 24; Accepted 2024 Mar 18
Copyright: © 2024 Neal et al.
Copyright year: 2024
Copyright holder: Neal et al.
License: This is an open access article distributed under the terms of the Creative Commons Attribution License, which permits unrestricted use, distribution, reproduction and adaptation in any medium and for any purpose provided that it is properly attributed. For attribution, the original author(s), title, publication source (PeerJ) and either DOI or URL of the article must be cited.
License URL: https://creativecommons.org/licenses/by/4.0/

Keywords: Trematode, division of labor, redia

Funding: National Institute of General Medical Sciences of the National Institutes of Health P20GM103449 NIGMS or NIH Norwich University Undergraduate Research Program Chase Student Endowment The research reported in this publication was supported by an Institutional Development Award (IDeA) from the National Institute of General Medical Sciences of the National Institutes of Health under Grant Number P20GM103449. Student authors were supported by the Norwich University Undergraduate Research Program, including funds from the Chase Student Endowment. The funders had no role in study design, data collection and analysis, decision to publish, or preparation of the manuscript.

==============================
Background

Prior research suggests that trematode rediae, a developmental stage of trematode parasites that reproduce clonally within a snail host, show evidence of division of labor (DOL). Single-species infections often have two morphologically distinct groups: small rediae, the ‘soldiers’, are active, aggressive, and do not appear to reproduce; large rediae, the ‘reproductives’, are larger, sluggish, and full of offspring. Most data supporting DOL come from trematodes infecting marine snails, while data from freshwater trematodes are more limited and generally do not supported DOL. The shorter lifespan typical of freshwater snails may partially explain this difference: defending a short-lived host at the expense of reproduction likely provides few advantages. Here, we present data from sixty-one colonies spanning twenty species of freshwater trematode exploring morphological and behavioral patterns commonly reported from marine trematodes believed to have DOL.

Methods

Trematode rediae were obtained from sixty-one infected snails collected in central Vermont, USA. A portion of the COI gene was sequenced to make tentative species identifications (‘COI species’). Samples of rediae were photographed, observed, and measured to look for DOL-associated patterns including a bimodal size distribution, absence of embryos in small rediae, and pronounced appendages and enlarged pharynges (mouthparts) in small rediae. Additional rediae were used to compare activity levels and likelihood to attack heterospecific trematodes in large vs. small rediae.

Results

Many of the tests for DOL-associated patterns showed mixed results, even among colonies of the same COI species. However, we note a few consistent patterns. First, small rediae of most colonies appeared capable of reproduction, and we saw no indication (admittedly based on a small sample size and possibly insufficient attack trial methodology) that small rediae were more active or aggressive. This differs from patterns reported from most marine trematodes. Second, the small rediae of most colonies had larger pharynges relative to their body size than large rediae, consistent with marine trematodes. We also observed that colonies of three sampled COI species appear to produce a group of large rediae that have distinctly large pharynges.

Conclusions

We conclude that these freshwater species likely do not have a group of specialized non-reproductive soldiers because small rediae of at least some colonies in almost every species do appear to produce embryos. We cannot rule out the possibility that small rediae act as a temporary soldier caste. We are intrigued by the presence of rediae with enlarged pharynges in some species and propose that they may serve an adaptive role, possibly similar to the defensive role of small ‘soldier’ rediae of marine trematodes. Large-pharynx rediae have been documented in other species previously, and we encourage future efforts to study these large-pharynx rediae.

Introduction

Division of labor (DOL) refers to the specialization of certain sets of individuals (cells, organisms, etc.) within a larger group to take on dedicated tasks, generally to increase the efficiency of the group as a whole. This includes reproductive DOL in which only a subset of individuals in the group reproduce. Well-known examples include the division of ant colonies into specialized castes, including workers and reproductive queens (summarized by Wilson, 1985, Robinson, 1992, and many others), but division of labor has also been documented in other arthropods, sea anemones and naked mole rats (reviewed by Hechinger, Wood & Kuris, 2011, Whyte, 2021). A decade ago, Hechinger, Wood & Kuris (2011) presented multiple lines of morphological and behavioral evidence from the trematode Himasthla sp. B suggesting that the intramolluscan stages of this parasite displayed reproductive division of labor, potentially adding a new taxonomic group to the relatively limited list of species with reproductive DOL.

Trematodes are flatworm parasites with complex, multi-host life cycles that typically include two or three hosts. Vertebrate animals including humans serve as definitive hosts for trematodes and are the site of sexual reproduction. Asexual reproduction occurs in the first intermediate host, which is usually a snail. Clonal proliferation in the snail produces a population of genetically identical individuals. The asexually reproducing trematode stages come in two main types: sporocysts, which are mainly comprised of germinal tissue and embryos contained in a sac-like body, and rediae, which are similar but additionally have a mouth, muscular pharynx, and gut. A single clonal group of trematodes can easily utilize all available space within its snail host, but they may face competition from other trematodes for space and resources during an infection. Evidence suggests that the community of trematode species that utilize the same host snail species interact with one another, even forming complex dominance hierarchies (e.g. Kuris, 1990; Kuris & Lafferty, 1994; Fernandez & Esch, 1991; Sousa, 1993). Redia-producing trematodes often have a competitive advantage in these hierarchies (Kuris, 1990; Sousa, 1992), and competitively dominant species may tend to evolve soldiers (Hechinger, Wood & Kuris, 2011).

Hechinger, Wood & Kuris (2011; and shortly thereafter Leung & Poulin, 2011) were the first to propose trematodes had division of labor when they demonstrated two distinct size classes of rediae and highlighted several differences between them. Large rediae are lethargic, plump and contain developing embryos and cercariae, a dispersive larval stage produced by sporocysts/rediae that leave the snail host to continue the parasite’s life cycle. Small rediae are active, slim, and free of developing embryos, but possess large (relative to their bodies) muscular pharynges that they use to attack heterospecifics at a much greater rate than the large rediae. The authors proposed that these small rediae forgo reproduction to act as soldiers, although the temporary or permanent nature of the caste remains unresolved (Mouritsen & Halvorsen, 2015; Garcia-Vedrenne et al., 2017). The small, thin size of these soldiers allows them to pass easily through snail tissues in search of would-be competitors (other trematodes), and their muscular pharynges allow them to consume or destroy these unwanted co-inhabitants. Since the publication of this article (Hechinger, Wood & Kuris, 2011), researchers have documented thirteen additional trematode species infecting marine/estuarine snails throughout the world that also show evidence of DOL (Table 1), and progress has been made toward understanding whether and when investment in soldier vs. reproductive rediae might be adjusted (reviewed by Poulin, Kamiya & Lagrue, 2019). Interestingly, despite substantial interest in this phenomenon (e.g., Newey & Keller, 2010; Poulin, Kamiya & Lagrue, 2019; Whyte, 2021), most work on trematode DOL has focused on marine/estuarine trematode species, with trematodes infecting freshwater snails receiving less attention (Table 1).

Table 1 Summary of data from previous studies investigating division of labor (DOL) in trematodes.

The first column (Species) indicates the trematode species from which data are reported. Columns 2–9 (Volume, Morphology, … Distribution) correspond to the patterns stated in Table 2. For all patterns, numeric values in the table are consistent with the predictions of DOL unless bolded and qualitative results generally indicate: Y-observations are consistent with DOL; N-observations are not consistent with DOL or no secondary morphs were observed for comparison. In the table, any result recorded in parentheses indicates results that were suggestive (e.g. observable in figures) but not specifically stated in the referenced article. Blanks indicate patterns that were not mentioned in the reference. The last two columns indicate the number of colonies (infected snails) sampled (Colonies) and the reference in which these data were recorded (Reference).

Species1	Volume2	Morphology3	Pharynx4	Activity5	Attacks6	Reproduction7	%Small8	Immature9	Distribution10	Colonies11	Reference12	
MARINE/ESTUARINE SPECIES	
Superfamily Echinostomatoidea , Family Himasthlidae	
Himasthla sp. B	63x	Y	20x	Y	0/100	Y	11–76	Y	Y	7–51	1	
Himasthla rhigedana	21x	Y	21x	Y	0.4/81	Y*	11–76	6.7	Y	5	2	
Himasthla elongata	B	N		Y	14/40	Y*/N	14–22			10, 18	3–4	
Acantho. spinulosum	39x	Y	26x	Y	1.6/35	Y*	0.94			5	2	
Acantho. sp. I	12x		25x	Y	Y	Y	17		Y	5–6	5	
Superfamily Echinostomatoidea , Family Philophthalmidae	
Philophthalmus sp.	B (Y)			Y	Y	Y*	43–77			30	6	
Cloa. michiganensis	141x	Y	40x	Y	0/31	Y*	3.6		Y	5	2	
Parorchis acanthus	216x	Y	114x	Y	0/12	Y*	3.8		Y	5	2	
Philophthalmid sp. I	65x		18x	Y	Y	Y	50		Y	9–10	5	
Philophthalmid sp. II	160x		45x	Y	Y	Y	75		Y	8–9	5	
Superfamily Opisthorchioidea , Family Heterophyidae	
Euha. californiensis	28x	NA	31x	Y	0/7.9	Y*	0.8		Y	4	7	
Phocitremoides ovale	69x	NA	71x	Y	N	Y*	0.2		Y	3	7	
Pygi. spindalis	32x	NA	31x	Y	N	Y*	1		Y	3	7	
Strictodora hancocki	19x	NA	21x	Y	1/30.8	Y*	2.9		Y	5	7	
FRESHWATER SPECIES	
Superfamily Echinostomatoidea , Family Echinostomatidae	
Echinostoma liei	N	(N)	N	N	N	N	N	17	(Y)	6	7	
Echino. spiniferum	N	N	(N)			N	15–28		Y	2–7	8	
Superfamily and Family Not Specified	
Unknown	N									18	9	
Notes:

1 Most genus names are fully written out except: Acantho.-Acanthoparyphium, Cloa.-Cloacatrema, Euha.-Euhaplorchis, Pygi.-Pygidiopsoides, Echino.-Echinostoma.

2 Is volume or standard linear size of rediae bimodal? Numbers indicate the number of times larger primary morphs are relative to secondary morphs with respect to volume; qualitative results: Y- bimodal and non-overlapping, B- bimodal (may not be non-overlapping), N- neither.

3 Do morphological structures (appendages, collar) differ between primary and secondary morphs? Y- structures are pronounced only in secondary morphs, N- structures present in both morphs or no secondary morphs observed, NA- structures present in neither morph.

4 Is pharynx volume is larger in secondary morphs? Numbers indicate the number of times larger the relative pharynx size of secondary morphs is compared with primary.

5 Are activity levels higher for secondary morphs?

6 Are attack rates higher for secondary morphs? Numbers are attack rates (%) on heterospecific trematodes of primary morphs/secondary morphs. Numbers are not directly comparable as some are per capita rates and some are per trial rates.

7 Are reproductive structures (germinal balls/embryos) absent from secondary morphs? Y-yes, Y*-yes, but this is how primary vs. secondary morphs were defined in this article, N-germinal balls were found in both morphs or no secondary morphs were observed.

8 What percentage of rediae are small (secondary morphs)?

9 What percentage of rediae are immature/intermediate?

10 Does the distribution of rediae within the snail host put secondary morphs/small rediae more commonly at invasion fronts (e.g., head/foot)?

11 Number of colonies sampled.

12 References are: 1-Hechinger, Wood & Kuris (2011), 2-Garcia-Vedrenne et al. (2016), 3-Nielsen, Johansen & Mouritsen, 2014, 4-Galaktionov et al. (2015), 5-Miura (2012), 6-Leung & Poulin (2011), 7-Garcia-Vedrenne et al. (2017), 8-Zikmundová (2011), 9-Hawkins & Bernot (2021).

Table 2 Summary of patterns commonly observed in trematode species for which division of labor has been suggested.

Table 1 summarizes which species have shown evidence of these patterns.

Pattern	Description	
Volume	The body volume or standard linear size (SLS is directly proportional to the cubed root of volume, Galaktionov et al., 2015) of primary vs. secondary morphs shows a bimodal and often non-overlapping pattern.	
Morphology	Morphological differences exist between primary and secondary morphs; specifically, secondary (and not primary) morphs have more pronounced collar and posterior appendages.	
Pharynx	The size (approximate volume) of the pharynx relative to the body is larger for secondary morphs than for primary morphs.	
Activity	Secondary rediae show greater activity or distance moved proportional to their body size over a short period of time.	
Attacks	Secondary morphs attack heterospecific trematodes at higher rate than primary morphs attack heterospecific trematodes	
Reproduction	Secondary morphs lack germinal balls/embryos even though nearly all primary morphs have germinal balls or developing embryos.	
%Small	The smaller secondary morphs make up a substantial percentage of the rediae in the infection.	
Immature	Immature rediae or rediae with phenotypes intermediate between primary and secondary morphs are rare.	
Distribution	Secondary morphs and immature rediae are more common than primary morphs/mature rediae in the snail’s mantle/mid-region/anterior.	

Freshwater trematodes have both key similarities and key differences when compared with marine/estuarine trematodes. One important similarity is that the intramolluscan stages of all trematodes live in clonal groups that share a localized, defendable resource (the snail), which favors the evolution of DOL (Hechinger, Wood & Kuris, 2011). Also, both freshwater and marine trematodes face invasion of the snail by other trematodes, and interspecific antagonism is well known among freshwater trematodes (Lie, Basch & Umathevy, 1965; Sousa, 1992; Kuris & Lafferty, 1994). The ability to directly attack one’s competitor provides an advantage that could be increased by a specialized class of soldier rediae (Poulin, Kamiya & Lagrue, 2019).

One important difference between marine and freshwater trematodes is the average longevity of their host snails. Freshwater snails tend to be short-lived in comparison with marine snails (Heller, 1990), and a short-lived host reduces the benefit of defending the snail host from potential competitors (Hechinger, Wood & Kuris, 2011; Poulin, Kamiya & Lagrue, 2019): if the host will die soon anyway, there may be less benefit of diverting resources to a specialized soldier class; it is likely more prudent to instead invest in quick production of dispersal stages. Hechinger, Wood & Kuris (2011) predicted that short-lived hosts may favor the development of “large, totipotent rediae” for defense; large rediae that are active and aggressive may retain the ability to reproduce while simultaneously defending against competitors; however, they are more likely to harm the host, but the cost of damaging the host is much less important in a short-lived host.

Because of these key similarities and differences, examining DOL-associated patterns (Table 2) in freshwater trematode species will provide important comparative data to help determine how widespread DOL is among trematodes, how DOL evolved in trematodes, the importance of certain selective pressures like snail host longevity for the evolution of DOL, and the utility of certain patterns in identifying trematode species that likely have DOL.

We therefore performed a preliminary survey of trematodes infecting freshwater snails in central Vermont, USA to determine whether any showed patterns consistent with patterns reported previously from marine trematodes. In Tables 1 and 2, we summarize some of the patterns most commonly recorded in previous studies, and we focus on these patterns in our data collection and analysis. We additionally attempted to monitor the development of small rediae in culture, which was mentioned by Lloyd & Poulin (2012).

Materials & methods

Collection and identification of infected snails

We collected snails by hand or net from four sites in central Vermont during the summer months (June–August) of 2019. The sites selected for this study were chosen based on the relatively high frequency of redia-producing infections during our collections in a previous summer (not all trematode species produce rediae- some only produce sporocysts). These sites are Waterbury Reservoir (WaR; Waterbury, Washington County), Ticklenaked Pond (TP; Ryegate, Caledonia County), North Montpelier Pond (NMP; Washington County) and Shelburne Pond (SP; Shelburne, Chittenden County), and were accessed via state-maintained swimming beaches (WaR) or boat launches (TP, NMP, SP). A few additional infected snails collected after the initial survey in fall 2019 and summer 2021 were included in the analysis opportunistically because we noticed that they had interesting features while sampling for another project.

Collected snails were identified to family using the key in Thorp & Covich (2010) and placed in individual containers of native water with constant illumination overnight to promote cercarial shedding. The water around each snail was scanned for cercariae with a dissecting microscope 1–2 days after collection to identify infected snails. Cercariae were photographed for reference and a sample was collected for DNA analysis. Cercariae for DNA analysis were added to ethanol (roughly 2/3 absolute ethanol, 1/3 water and cercariae to achieve roughly 70% ethanol), allowed to settle, and had excess liquid removed before being stored frozen. If it was not possible to collect enough cercariae (>5–10) for DNA analysis, rediae were collected following dissection and also stored frozen in a small amount of ~70% ethanol. The 70% ethanol was allowed to evaporate just prior to DNA extraction.

Infections that produced cercarial types consistent with redia-producing trematode species were housed in larger individual cups with native or distilled water until they could be dissected. Infections producing cercarial types not consistent with redia-producing trematodes were dissected to confirm the presence of sporocysts and ensure no redia-producing infections were missed. Some of these non-redia infections were used as heterospecifics in the Attack Tests (below).

Species identification

Trematode species were identified using DNA barcoding of the Folmer region of COI. DNA was extracted from stored cercaria (or occasionally rediae) samples after any remaining ethanol was allowed to evaporate using an Omega EZNA Tissue kit. DNA was amplified using Dice1/11 or Dice 1/14 primer pairs (Van Steenkiste et al., 2015). Sanger Sequencing was performed at an external facility (Yale University’s W. M. Keck Biotechnology Research Laboratory or Eurofins: eurofins.com). Forward and reverse trace files were aligned and viewed using SeqTrace (Stucky, 2012) and unreliable base calls were removed. Removal of unreliable bases include trimming ends where quality scores were not consistently above 20–30, examining and editing any mismatched calls between the forward and reverse sequence, and removing (setting to ‘N’) base calls from any site with apparent polymorphisms (these were very rare). Sequences from all samples were aligned using Clustal Omega (McWilliam et al., 2013) and the resulting percent identity (PI) matrix was used to calculate sequence divergence; sequences with <5% divergence were grouped into genetic groups that likely correspond to species (termed ‘COI species’ here; Gordy et al., 2016 and others use a similar approach). One sample from each COI species was then entered into BLAST (Altschul et al., 1990) for comparison with the NCBI database to determine a tentative species identification for each COI species.

Isolation of rediae

Redia-infected snails were crushed gently and removed from their shells, then separated into three regions: apical (gonad/digestive), middle, and head/foot. Each body section was teased apart in a separate Petri dish to isolate the rediae. Most snails were dissected in distilled water, but toward the end of the summer 2019 we discovered that dissecting in dilute saline (0.1% as suggested in Schell, 1970) substantially increased the activity levels of rediae, so dilute saline was used for dissections from that time on (starting August 1, 2019). For some infections, body sections were not separated for one of several reasons including 1) the infection was not discovered until after the snail was partially dissected or 2) the snail was damaged while crushing the shell, so it was not possible to adequately separate body sections.

Morphological measurements

Once the snail sections had been teased apart, we collected a sample of rediae for morphological analysis by pipetting a sample of water from the dissected material with all rediae in a haphazardly selected area of the Petri dish onto a microscope slide. This was performed without observing what size rediae were being collected, but when individual rediae were collected with a pipette it did not seem that any size redia were more difficult to collect, so this method should be relatively unbiased. We sampled a target of 100 rediae per section, and for any body section with fewer than 100 rediae, the snail tissue and surrounding water were examined carefully to locate as many rediae as possible.

Once the rediae were collected, each redia was photographed for measurement. To facilitate quicker data collection, rediae were not fixed or otherwise immobilized, but most that were dissected in water were not very active and the noise introduced by redial movement is likely dwarfed by the large differences in actual body size. Measurements were taken from photos using Spot Basic software (spotimaging.com/software/spot-basic/; by MS) or ImageJ (Rasband, 1997–2018; by ATN); measurements for a single infection were always made by the same researcher/software package and a subset of measurements that were repeated by the other researcher/software package did not differ substantially. Measurements taken were: (1) length, using a segmented line down the midline of the redia, (2) width, measured at the widest point (excluding obvious protrusions like collars and appendages) and (3) pharynx diameter (longest dimension; most were roughly circular in photos).

After measurements were complete, rediae were sorted by size and we examined the photos of the smallest five and largest five rediae sampled per infection for evidence of an anterior appendage/collar and posterior appendages (to assess differences in morphology) and cercaria/embryos/germinal balls (to assess reproductive potential). Our ability to identify these structures depend on the quality of the photo (including the position of the redia in the photo), so it was not always possible to definitively determine the presence or absence of every structure. Instead, presence was scored as 0 (definitely not present), one (probably not present), two (unclear), three (probably present), or four (definitely present). We refer to this below as the ‘presence score’ for each trait.

Activity measurements

For colonies with sufficient rediae, we selected the ten largest and ten smallest rediae remaining from the dissection after the haphazard morphology sample was removed. Each redia was isolated in dissection medium (water or dilute saline) in its own well in a 48-well cell culture plate. Photos were taken 2 s apart immediately after rediae were placed in the wells and again after 15 min. When it was not possible to fit the whole redia in one microscope field, only the anterior end of the redia was photographed. Photo pairs were superimposed using ImageJ and movement distance was recorded by measuring the distance between the same point on the redia in the two successive photos. Whichever part of the anterior end of the redia that travelled the furthest during the 2 s interval was used for measurement (File S1). We also used ImageJ to measure the total length of each redia.

Attack tests

Again for colonies with sufficient rediae, we assessed the tendency of smaller vs. larger rediae to attack colony mates, conspecifics and heterospecific trematodes. For this test, we selected the 20–30 largest and 20–30 smallest rediae remaining after the previous two tests and split them into three cell culture plate wells per redia size. The first ten rediae of each size were combined with an additional ten rediae of any size from the same infection (‘colony mates’). The next ten rediae of each size were combined with ten rediae from another colony of the same species (when available; species based on morphology; ‘conspecifics’). The final ten rediae of each size were combined with ten sporocysts from a colony of a different species (‘heterospecifics’). After allowing the trematodes to interact in the well, each well was observed for at least 30 s to determine whether any rediae were attacking (attached to other rediae or sporocysts by their mouthparts). We did not standardize or record the amount of time trematodes interacted prior to being observed, but recall it being around thirty to sixty minutes.

For most colonies, several researchers worked at two camera-equipped microscopes simultaneously to collect data on morphology, activity and attacks, meaning that even though the rediae for the attack tests were selected after those for the morphological measurements and activity tests, the delay in collecting these data was not great. The total time between snail dissection and the completion of photography (for all purposes) was typically less than 3 h.

Size designations

Rediae were assigned to a size category (‘small’ vs. ‘large’) in one of two ways; for infections with a bimodal distribution of volume (see below), we identified the break (bin with fewest observations) between peaks on the histogram and used this as a cutoff for size category. Histogram bin sizes were the default in R’s hist() function and varied by sample size. For infections without a bimodal size distribution, we compared the rediae in the smallest 25% of volumes to those in the largest 25%. This allowed us to compare small and large rediae even if they do not appear to form two distinct groups.

Culture

We attempted to maintain rediae in culture for several weeks to monitor the development of small rediae. Prior to dissection, the snail shell was wiped with water to remove visible dirt, wiped twice with 70% ethanol to disinfect, and then soaked for approximately 5 min in a 0.1% saline with 100 μg/mL gentamycin (“antibiotic saline”). The snail was dissected in antibiotic saline in a sterile petri dish in a laminar flow hood. Rediae were rinsed three times in 1,000 μL antibiotic saline. 100 μL of rediae (about 7–22 rediae) were then transferred to each of 12 wells of a 24-well cell culture plate. Because we were not sure what conditions would best promote redial survival and growth, the wells contained one of six different treatments (two replicates of each): (1) redia co-cultured with snail cells (Biomphalaria galbrata embryonic cells; Bge) in a 1:1 mixture of Bge Medium and Medium F (described below), (2) redia co-cultured with snail cells in a 1:3 Bge:F media, (3) redia co-cultured with snail cells in 100% Bge Medium, (4) redia alone in a 1:1 mixture of Bge:F media, (5) redia alone in a 3:1 mixture of Bge:F media, and (6) redia alone in 100% Bge Medium. Rediae were checked every 1–2 weeks for 11 weeks (77 days) to assess survival and growth. Survival was assessed by watching each redia for 5 s. If any motion was detected, it was considered active (and alive). Growth was assessed by photographing small rediae born in culture and measuring their body length. For small rediae that were active, multiple photos were taken to reflect both their stretched and contracted length.

The snail cells used in this study—Bge Cell Line, NR-40248—were provided by the NIAID Schistosomiasis Resource Center distributed through BEI Resources, NIAID, NIH (https://www.beiresources.org/Catalog/cellBanks/NR-40248.aspx; Taus et al., 2013). Prior to inclusion in this study, Bge cells were cultured in Bge Medium using the formulation provided with the cells. Co-culture with Bge cells has been shown to promote the longevity and development of a variety of freshwater trematode species (reviewed in Coustau & Yoshino, 2000). Medium F was shown to promote redial survival in a study with marine trematodes (Lloyd & Poulin, 2011), and the recipe for Medium F can be found in the supplemental materials of that article.

Analysis

All analysis was performed in R version 4.2.1 (R Core Team, 2022). Unreliable measurements, for example from damaged rediae, were removed prior to analysis. For most analyses, all rediae from a single infected snail were pooled for analysis. Analysis was performed on each infection individually; this allowed us to easily determine how consistent patterns were among infections of the same genetic group. We did not adjust our p-value cutoff for multiple comparisons (we used α = 0.05), so we expect about 5% of ‘statistically significant’ results to be false positives. Below we describe how we tested for each DOL-associated pattern.

Bimodal Volume: to test whether redia volumes showed a bimodal distribution, we first estimated the volume of each redia assuming its shape is roughly cylindrical. We then performed a Shapiro-Wilk Normality Test on the log-transformed volume. For infections that differed significantly from a normal distribution, we examined histograms of the log(volume) to determine whether there were two distinct peaks.

Appendages: to test whether smaller rediae have more pronounced appendages (anterior appendage/collar and posterior appendages) than larger rediae, we used a Wilcoxon Rank Sum Test to test the alternative hypothesis that the smallest five rediae had a higher average presence score for each appendage than the largest five redia from each infection.

Pharynx: to test whether smaller rediae have larger pharynges relative to their body size, we first estimated the volume of each pharynx assuming its shape is roughly spherical and divided pharynx volume by redia volume to obtain a relative pharynx volume. We compared relative pharynx volume between large and small rediae using a one-tailed Wilcoxon Rank Sum test. We also compared absolute pharynx volume in the same way.

Activity: before comparing the distance moved in two seconds by small vs. large rediae, we first checked that the rediae selected for analysis were indeed ‘small’ and ‘large’ in the context of all rediae measured (for a few infections, we failed to select the smallest rediae because they were rare or less obvious under the dissecting microscope). We re-classified any redia sizes that were not internally consistent (e.g. any ‘small’ redia that exceeded the length of any ‘large rediae) or not consistent with the size categories established for other analyses (see ‘Size designations’ above). We then averaged the measurements made at 0 and 15 min and compared the relative distance moved (distance moved/body length) for small vs. large rediae for each infected snail using the Wilcoxon Rank Sum Test.

Reproductive potential: we recorded the maximum presence score for embryos for the smallest five rediae in each infection.

Distribution of morphs in different body sections: we calculated the proportion (with 95% confidence interval) of small and large rediae (intermediates excluded) found in each body section that were designated ‘small’. We compared these proportions among body sections to determine whether there was a higher proportion of small redia in the anterior regions of the snail, especially the head/foot (or mid section if no rediae were found in the head/foot).

Results

In summer 2019, we collected 3,496 snails from six families (File S2), of which 292 were infected (File S3). Of these, we were able to collect data on rediae from 59 infections. The remainder of the infections either did not produce redia (many trematode species only produce sporocysts) or did not survive long enough in the lab for data collection (less common). An additional seven infections that were collected for another project in 2021 were added later because their rediae seemed to have interesting features (see ‘Additional observations’, below). In total, we measured length, width or pharynx size for 7,546 rediae. Prior to analysis, we excluded any measurements that were deemed unreliable by the measurer; the most common reason for exclusion was the redia being damaged or obscured in the photo. We also excluded data from three infections that appeared to have issues with the numbering or labeling of photos and one infection that appeared to be a mixed infection with two redia-producing species (details in File S4).

Two discrepancies in measurements between MS and ATN were noted after measurements were complete; first, ATN’s measurements were consistently about 5% higher than MS’s, likely due to a slight difference in calibration of the software on each investigator’s computer. Second, MS always measured pharynx size perpendicular to body length rather than measuring the longest dimension. Neither discrepancy should substantially affect measurements overall, nor should they impact study conclusions because all comparisons were all made among rediae within an infection and no infection’s data combined measurements from the two investigators.

We were able to successfully sequence a region of COI for 59 of the 66 trematode infections collected (54 of the 61 infections with included size data). Grouping infections with at least 95% sequence similarity, we ended up with 20 COI species, each with 1 to 16 infections (Table 3). Some of these COI species matched (>95 percent identity) sequences contained in the NCBI database, while others did not (Table 3). File S6 shows representative photos of cercariae and rediae for each of the COI species.

Table 3 Summary of infections analyzed in this study with nearest match in the NCBI database.

Each row corresponds to a purported species (‘COI species’) that was identified by sequencing COI and grouping infections with at least 95% matching nucleotides. The sequence for one sample from each COI species was used to identify the nearest match in the NCBI database along with the percent identity (<95% is likely a different species; NCBI (PI)). The table also lists the superfamily the matching species in NCBI belongs to (Superfamily), the number of colonies (infected snails) we sampled that belong to each COI species (N), and the GenBank accession number for our sequencing results for a representative member of the COI species.

COI Species1	NCBI (PI)	Superfamily	N2	GenBank3	
H107	Paragonimus mexicanus (77%)	Allocreadoidea	1	OR666891	
P113	Clinostomum marginatum (99.8%)	Schistosomatoidea	6	OR666897	
H070	Faciola gigantica (79%)	Echinostomatoidea	2	OR666887	
H102	Echinochasmidae sp. isolate 2 (83%)	Echinostomatoidea	1	OR666889	
P107	Echinostoma trivolvis comp. sp. Lin. A (99.8%)	Echinostomatoidea	2	OR666895	
P112*	Psilostomidae gen. sp. A (93%)	Echinostomatoidea	16	OR666896	
P115	Echinostoma trivolvis (99.1%)	Echinostomatoidea	5	OR666898	
P150	Echinostoma trivolvis (87%)	Echinostomatoidea	1	OR666914	
P153	Drepanocephalus auratus (100%)	Echinostomatoidea	1	OR666916	
P177	Echinoparyphium sp. Lineage 3 (91%)	Echinostomatoidea	2	OR666927	
Ph096	Echinostomatidae sp. (100%)	Echinostomatoidea	1	OR666930	
Ph116	Echinoparyphium sp. A (98%)	Echinostomatoidea	1	OR666936	
Ph159	Echinoparyphium sp. A (87%)	Echinostomatoidea	1	OR666938	
V141	Echinostoma bolschewense (91%)	Echinostomatoidea	6	OR666939	
V172	Echinoparyphium sp. E (74%)	Echinostomatoidea	1	OR666942	
H127	Quinqueserialis quinqueserialis (75%)	Notocotyloidea	1	OR666893	
Ph095	Notocotylus sp. A (99.7%)	Notocotyloidea	4	OR666929	
H104	Amphimerus (78%)	Opisthorchioidea	1	OR666890	
P141	Zygocotyle lunata (87%)	Paramphistomoidea	1	OR666913	
Ph100	Plagiorchis (80%)	Plagiorchioidea	1	OR666931	
Unknown	Unknown	Unknown	6	(NA)	
All			61		
Notes:

1 COI species corresponds to an alphanumeric code assigned to the first infection collected from each species (assigned based on clustering of COI sequences). The letters in the code correspond to the family of snail from which the infection was collected, where H = Hydrobiidae, Ph = Physidae, P = Planorbidae and V = Viviparidae. For the most part, COI species were not found in more than one family of snail. The starred group (P112) may include two closely related groups; some percent identity values on COI sequences for this group were 92–95%, but it would have been difficult to conclusively separate the group into two groups.

2 These numbers include only infections with size measurements included in the study.

3 GenBank Accession Numbers for a representative sample from this COI species (other samples in COI species share at least 95% of the sequence except where noted; accession numbers for all samples are available in File S4).

One issue that will affect the results somewhat is the osmotic stress placed on trematodes dissected from their snail host in water rather than saline during the first part of summer 2019. One sampled COI species (P112) included a reasonable number of infections dissected in both water (N = 8) and saline (N = 8); pooling infections, water-dissected trematodes show a 54% increase in median volume when compared with saline-dissected trematodes. That said, we noted no consistent differences when comparing the qualitative morphological results obtained from saline-dissected vs. water-dissected infections for three COI species (P112; P113 with four water and two saline infections; P115 with four water and one saline infection; Table 4) and we note that while substantial, a 54% increase in volume is still relatively small compared with the 10s to 100s fold difference in size for some redial morphs (Table 1), so we have retained the morphological data from water-dissected infections but reported the results separately in Table 4. Conversely, dissection medium seemed to have a large impact on behavior, so only results from saline dissections are reported.

Table 4 Summary of results from this study investigating patterns linked to division of labor (DOL) in trematodes from freshwater snails.

The number of colonies displaying each pattern (Volume, Reproduction… Attacks) for every COI species are indicated by the numbers in the pattern columns. The COI species column also indicates the dissection medium (water-w- vs. saline-s) in parentheses. Like Table 1, results inconsistent with DOL are in bold.

COI species1	Volume2	Reproduction3	Morphology4	Pharynx5	%Small6	Distribution7	Activity8	Attacks9	
Superfamily Allocreadioidea	
H107 (w)	Y	1Y?	1N	1Y	52%	1O			
Superfamily Clinostomatoidea	
P113 (w)	4Y	3U/1Y?	4N	4Y	14–43%	2Y/1O/1NA			
P113 (s)	1Y/1N*	2U	2N	2Y	15%	2NA			
Superfamily Echinostomatoidea	
H070 (w)	2N	1U/1Y	2N	1Y/1N	NA	1O/1NA			
H102 (w)	1N	1Y	1N	1NA	NA	1O			
P107 (w)	1N*/1Y	2Y	2N	2Y	12%	1Y/1NA			
P112 (w)	5Y/3N*	2U/4Y?/2Y	1Y/2O/5N	7Y/1N	4–13%	2Y/6O			
P112 (s)	1Y/7N*	2Y?/6Y	2O/6N	3Y/4N/1NA	24%	8O	7NS		
P115 (w)	1Y/3N*	2Y?/2Y	4N	4Y	3%	1Y/1O/1N/1NA			
P115 (s)	1Y	1U	1N	1Y	12%	1NA			
P150 (w)	1Y	1Y	1N	1Y	7%	1O			
P153 (w)	1Y	1Y?	1O	1Y	14%	1O			
Ph096 (w)	1N*	1Y?	1N	1Y	NA	1NA			
Ph116 (w)	1Y	1Y?	1Y	1Y	18%	1O			
Ph159 (s)	1N*	1Y	1N	1Y	NA	1NA			
V141 (w)	1N*/1N	1Y?/1Y	2N	2Y	NA	2NA			
Superfamily Notocotyloidea (Pronocephaloidea)	
H127 (w)	1Y	1U	1N	1Y	50%	1O			
Ph095 (w)	1Y/2N*/1N	1U/3Y?	1Y/3N	3Y/1N	8%	1Y/2O/1NA			
Superfamily Opisthorchioidea									
H104 (w)	1N	1Y	1N	1NA	NA	1NA			
Superfamily Paramphistomoidea	
P141 (w)	1N*	1Y	1N	1Y	NA	1O	1NS	1N	
Superfamily Plagiochioidea	
Ph100 (w)	1N*	1U	1N	1N	NA	1NA			
Unknown (no sequencing data)	
H096 (w)	1N*	1Y	1N	1NA		1NA			
H148 (w)	1Y	1U	1N	1Y	56%	1NA			
H158 (w)	1N	1Y	1N	1Y		1NA			
P135 (w)	1N*	1U	1N	1Y		1O			
P139 (w)	1N*	1Y	1Y	1Y		1NA			
Ph115 (w)	1N*	1U	1N	1Y		1NA			
Superfamily Echinostomatoidea (2021 Infections)	
P177 (s)	2Y	2U	1O/1N	2Y	5–17%	2NA			
V141 (s)	4N*	4Y?	4N	4Y	4NA	4NA			
V172 (s)	1N*	1U	1 N	1Y	1 NA	1NA			
All	23Y/31N*/7N	18U/22Y?/22Y	4Y/6O/51N	47Y/10N	3–56%	7Y/27O/1N	8NS	1N	
(Water only)	18Y/17N*/7N	12U/15Y?/15Y	4Y/3O/35N	35Y/4N	3–56%	7Y/19O/1N			
(Saline only)	5Y/14N*/0N	6U/7Y?/7Y	0Y/3O/16N	14Y/4N	5–24%	0Y/8O/0N	8NS	1N	
Notes:

1 Presumed species based on grouping of COI sequences by similarity. The species name is an alphanumeric code in which the letter indicates the snail species from which the first specimen was collected. The (w) or (s) in parentheses indicates whether the snails were dissected in water or saline; note that a few species are listed on two lines to separate results from different dissection media.

2 Is the volume of rediae bimodal? Y = appears bimodal; N* = does not appear bimodal, but differs from a normal distribution; N = does not differ from a normal distribution.

3 Are reproductive structures (germinal balls/embryos) present in small rediae? Y = germinal balls/embryos clearly seen in smallest 5 rediae (score = 4); Y? = germinal balls/embryos are probably in the smallest five rediae (but somewhat unclear in the photo; score = 3); U = germinal balls/embryos may or may not be present in smallest five rediae (unclear in photo; score = 2).

4 Do morphological structures (appendages, collar) differ between small and large rediae? Y = small rediae have more prominent anterior and posterior appendages (and appendage scores differ significantly); O = they differ for one appendage only (anterior or posterior); N = they do not differ significantly.

5 Is pharynx size larger in small rediae? Y = small rediae have significantly larger relative pharynx size than large; N = no significant difference; NA = insufficient data (at least one redia size has no relative pharynx measurements).

6 What percentage of rediae are small? Only infections with bimodal distributions are included in the analysis (others are marked NA). For genetic groups with more than one infection tested (number of infections corresponds to number of Y in Vol. column), a range is provided.

7 Does the distribution of rediae make small rediae more common at invasion fronts? Y = the ratio of small to large rediae in the snail’s head/foot (or mid section, if no rediae in foot) is higher than in the gonad (the confidence intervals do not overlap); O = confidence intervals on these ratios (proportions) overlap; N = ratio is larger in gonad; NA = data from multiple body sections not available.

8 Are small rediae more active? NS = no significant difference in the distance moved over a 2 s interval between small and large rediae.

9 Are small rediae more likely to attack? N = the number of observed attacks by small rediae did not outnumber the observed attacks by large rediae.

Body volume

The results examining the distribution of rediae volumes were somewhat mixed. Even though redia volumes differed from a normal distribution for 54/61 infections, the distribution of volumes in only about 23 infections appeared noticeably bimodal (Fig. 1, Table 4), and none showed the substantial gap in volumes between small and large redia that is more typical of the marine trematodes. Patterns within superfamilies and even within COI species were also quite variable.

Figure 1 The distribution of redial volumes for rediae from three infected snails included in this study.

Panels A and B were classified as bimodal in Table 4. Panel C was classified as not bimodal, but it does differ from a normal distribution.

Reproductive potential of small rediae

Because we did not conduct a detailed morphological assessment of small rediae (e.g. by fixing and staining them), it was not always clear from our photos whether each individual small redia sampled contained germinal balls or embryos. Nonetheless, we were at least relatively sure (presence score 3–4) that embryos were visible in the smallest five rediae in 43/61 infections, and for none of the infections were we confident that at least one of the smallest five rediae did not have the potential to reproduce (Table 4). For many small rediae, embryos were easily observable (e.g. Fig. 2).

Figure 2 Photos of small and large rediae for two trematode species included in this study.

For all pictured rediae, the anterior end of the redia is on the right side of the photo. (A) Small (inset) and large (main photo) rediae from an infection of the P152 COI species (see Table 3). Note the presence of embryos, collars (less pronounced in large redia) and posterior appendages in both small and large rediae. The pharynx is hard to see in the largest redia, but the pharynges are similar in size for all three visible rediae. (B) Small (inset) and large (main photo) rediae from an infection of the P141 COI species (see Table 3). Note the presence of embryos and absence of appendages in both small and large rediae. The pharynges are also similar in size (though a bit harder to see for the larger redia).

Appendages

For 50/61 infections, anterior and posterior appendages were not more prominent on small rediae than on large rediae either because both small and large rediae had appendages (e.g. Fig. 2A) or because neither did (e.g. Fig. 2B). The few infections that do show an apparent difference do not appear concentrated within a single superfamily or COI species.

Pharynx size

For 47/61 infections, relative pharynx size was larger in small rediae than large rediae (Table 4). For the remaining infections, sample size may have contributed to an inability to detect a difference: four infections did not have enough data to run the statistical test, and of the ten infections that did not show a statistically significant difference in relative pharynx size, nine had five or fewer pharynx measurements for at least one redia size (Files S4 and S5). Absolute pharynx size was generally a little larger in large rediae (significantly so for 35/61 infections; Files S4 and S5). Pharynx sizes in some example rediae can be seen in Fig. 2.

Distribution of rediae in snail

Only seven of the 37 infections for which rediae were separated by body region showed a significantly higher proportion of small rediae in the foot (Table 4). For fifteen of the remaining infections, the proportion of small rediae tended to be higher in the anterior regions of the snail, but confidence intervals overlapped.

Activity

We recorded data on the activity levels of rediae from 28 infections; 8/28 were dissected in dilute saline and the remainder were dissected in water. The distance that water-dissected rediae moved in 2 s was substantially lower than the distance moved by saline-dissected rediae in the same time: most (75%) water-dissected rediae moved less than 7 μm, while most (75%) saline-dissected rediae moved more than 13 μm. Because of this substantial discrepancy, we decided to exclude all measurements from water-dissected rediae, leaving only eight infections for analysis. Of these eight infections, most showed a tendency for large rediae to move further in 2 s than small rediae (Fig. 3A, p < 0.05 for 2/8 infections), but there did not seem to be a consistent difference in the distance moved for small and large rediae relative to their body size (p > 0.10 for all infections, Fig. 3B).

Figure 3 Comparison of distance moved by small vs. large rediae over 2 s.

(A) The absolute distance moved. (B) The distance moved relative to the redia length.

Attacks

We recorded the frequency of attacks by large vs. small rediae for only two infections dissected in dilute saline (and another twenty dissected in water). We observed attacks in one of the saline-dissected infections, with two attacks by large rediae and one by a small redia. Attacks were on a same-colony redia, a heterospecific cercaria and a heterospecific sporocyst (Table 4). We also observed attacks in two of the twenty water-dissected infections (not reported in Table 4 due to impact of osmotic stress). In one infection, we observed only a single attack by a large redia on another damaged redia from the same infection; it was unclear whether the attacking redia caused the damage or not. In a second water-dissected infection, we observed two large rediae and no small rediae attacking. Both attacks were on heterospecific sporocysts.

Culture

We monitored the rediae from only a single infection in culture, but decided to report our findings nonetheless. Rediae were taken from a snail in the family Planorbidae collected from NMP in October 2019. This infection is not included in Tables 3 and 4 because the rediae were not tested for any of the other patterns in this study (no volume distribution, pharynx size, activity, etc.); the rediae’s COI sequence was 96–98% similar to the P177 COI species from Table 3.

Rediae appeared to have higher survival when co-cultured with snail cells: by day 33, none of the approximately 84 rediae in the six wells without snail cells remained active, while all six wells with snail cells had at least some active rediae (7–35% of rediae per well, 88 rediae total). Four of the six wells with snail cells still had active rediae after 77 days (7–30% of rediae per well), after which we stopped recording data. It was unclear from our results whether the culture media had any significant influence on redia survival (File S7B).

In addition to the “adult” rediae that were initially added to each well, small “baby” rediae appeared in four of the wells (one with snail cells, three without). The number of small rediae per well increased over time for most wells (three increasing to six small rediae in the well containing snail cells; two increasing to three in two of the wells without snails cells; no increase in the final well without snail cells; File S7C). Like the larger rediae, the small rediae in wells without snail cells all ceased activity after 21 days (and some much earlier), but the small rediae in the well with snail cells all remained active throughout the 77 days over which they were monitored. We were unable to differentiate between the small rediae in each well to allow tracking of individual rediae sizes over time, but there is no apparent change in size of the small rediae when tracked collectively over 10 weeks (day 6, when they first appeared, to day 77, when we stopped keeping records; Fig. 4).

Figure 4 Size of small rediae over time in culture.

Data were collected at seven times; at each collection time, each point represents measurements from a single redia. The rediae were cultured in four wells of a cell culture plate; one well (circles) also contained cultured snail cells, which seems to improve redia survival. Redia lengths were measured from photos of rediae. For rediae that were actively moving when the photo was taken, multiple photos were captured and multiple length measurements were taken. The plot shows the mean (points) and range (vertical lines) of these measurements. Linear models fit to the mean redia sizes for each well did not show a significant change in redia length over time.

Additional observations

For a few infections collected in summer 2019 and several infections collected either fall 2019 or summer 2021, we observed one or more rediae with notably large pharynges compared with the other rediae in the infection (Fig. 5). We struggled to define exactly what constitutes a ‘large’ pharynx, but large pharynges were almost always identifiable as outliers (>[Q3 + 1.5*IQR]) from the distribution of pharynx sizes. Some infections without notably large-pharynx rediae did sometimes have one or a few rediae whose pharynx sizes were outliers, but pharynx-size outliers were more common in infections with notably large-pharynx rediae. For infections with large-pharynx redia (based on a visual inspection of photos), 5–15% of rediae had outlier pharynx sizes.

Figure 5 Photos of rediae with small vs. large pharynges from four separate infections.

Rediae in (A) are an infection of COI species P153 (see Table 3). (B and C) From COI species P177. (D) From COI species V141.

The large-pharynx rediae were not noticeably different in size from other large rediae (Fig. 6), though for some infections they did tend to be a bit wider relative to their length (e.g., Fig. 6D) and at least some appeared capable of reproducing (developing cercariae were observed in several large-pharynx rediae). The large-pharynx rediae often appeared to have a larger gut and darker body coloration than standard-pharynx rediae (Fig. 5). Large-pharynx rediae were noted in eight infections of three COI species: P153 (one infection of one sampled), P177 (two infections of two sampled), and V141 (five infections of six sampled).

Figure 6 Comparison of body dimensions for rediae with small (gray) vs. large (black) pharynges.

Each graph shows the rediae collected from a single infection. Photos of rediae from each infection are in Fig. 5. Rediae in (A) are an infection of COI species P153 (see Table 3) and correspond to the photo in Fig. 5A. (B and C) From COI species P177 and correspond to the photos in Figs. 5B and 5C. (D) From COI species V141 and correspond to the photos if Fig. 5D.

Discussion

Based on our preliminary analysis of over sixty trematode colonies representing twenty or more species, we do not believe that any of the sampled colonies demonstrate strong evidence supporting reproductive DOL, a finding consistent with prior data from freshwater trematodes (Table 1). We were at least moderately confident that small rediae contained embryos in 70% of the sampled colonies, and in no colonies were embryos clearly missing from the smallest rediae; if small rediae can reproduce, they do not represent a separate non-reproductive caste. This does not preclude the possibility that they carry out age-structured division of labor (Poulin, Kamiya & Lagrue, 2019), though we saw no evidence that small rediae were more active or aggressive in the (admittedly limited number of) colonies sampled. Nonetheless, our study highlights a number patterns and potential avenues for future research that may prove fruitful.

First, our study may provide important context for other studies on DOL. As Galaktionov et al. (2015) point out, several of the patterns that have been touted as evidence in favor of reproductive DOL have potential alternative explanations. By comparing these patterns between colonies or species that do and do not appear to have reproductive DOL, it may become more apparent which patterns are unique and which may be common to all or many trematode species, regardless of whether they have reproductive DOL. Such a comparison may help researchers understand the potential importance of certain redial traits to DOL or trematode biology in general.

One example is the relative pharynx size of large vs. small rediae. Even though the colonies we sampled do not appear to have reproductive DOL, the majority showed a larger relative pharynx size in small vs. large rediae similar to what is seen in marine species; like some marine species, the absolute pharynx size was not larger in small rediae. Pharynges may be relatively larger in small rediae simply because they do not grow as much as the rest of the body during development. That said, this may also predispose small rediae to being good soldiers in species for which DOL is favored.

The bimodal distribution of redial volumes may also not be indicative of DOL as it was relatively common among the infections we sampled; Galaktionov et al. (2015) provide an alternative explanation for this pattern based on development, though it is also worth noting that the wide separation between histogram peaks seen in marine trematodes (e.g., Garcia-Vedrenne et al., 2016 and 2017) was not observed in our data even for colonies that had some bimodality. Prior studies of DOL in freshwater trematodes have not shown redial volume to be bimodally distributed (Garcia-Vedrenne et al., 2017, Zikmundová, 2011, Hawkins & Bernot, 2021).

Our results also suggest several patterns that may distinguish trematodes with DOL. Differences in appendage prominence, activity levels and likelihood to attack between small and large rediae do seem exclusive to marine trematodes (especially the latter two, though our data testing those patterns is admittedly very limited), which might indicate that these patterns are more closely tied to DOL.

In addition to providing context for patterns associated with DOL, our study also presents an interesting direction for future exploration. For several infections, we noted the presence of a small group of rediae with unusually large pharynges. We do not believe these represent a coinfecting species due to the consistency with which they appear in the infections of certain species (two of two sampled infections for one species, five of six sampled infections for another). These large-pharynx rediae seemed to share several other morphological traits, such as a larger gut and more pronounced collar, but we were limited in our ability to fully describe these differences due to their relatively scarcity and the fact that we were working on a different study when we stumbled upon them. We attempted to sample additional infections with large-pharynx rediae during a short sampling period in 2022 to obtain more detailed data on the differences between small- and large-pharynx rediae, but were unable to locate additional infections, possibly due to temporal variation in the distribution/prevalence of the trematode species.

There are other reports of rediae with unusually large pharynges, such as the precocious mother redia described by Sapp, Meyer & Loker (1998), and mother and daughter rediae are known to differ in morphology for various species (e.g., Odening, 1965, cited by Galaktionov & Dobrovolskij, 2003; Kuntz & Chandler, 1956; stated more generally by Galaktionov & Dobrovolskij, 2003 p. 53). Mother rediae are likely to be relatively rare (esp. the precocious mother redia, of which there is only one per infection (Sapp, Meyer & Loker, 1998), so we do not believe the large-pharynx rediae we have observed, which appear to comprise up to 15% of rediae in the infection, are simply mother rediae. Additionally, many mother rediae produce primarily or exclusively rediae, while we have observed that some of these large-pharynx rediae contain cercariae. Many of the trematodes we collected likely belong to families with multiple redial generations (Table 3, Galaktionov & Dobrovolskij, 2003), and several prior studies have also documented the presence of large and small pharynx rediae of the same redial generation (Lie & Basch, 1967; Lie & Umathevy, 1965; Donges, 1963 cited in Lie & Basch, 1967). Lie & Basch (1967) note that the large-pharynx rediae have an enlarged gut, which we also observed.

Whether or not these large-pharynx rediae represent a different redial generation, their distinctive morphology suggests that they may play an adaptive role in the infection. Many questions remain, such as how consistently these large-pharynx rediae appear in infections of the same species, whether they appear in other closely or distantly related species (marine and freshwater), what role they play in the infections, and whether their relative frequency within a colony changes over the course of an infection or in response to environmental factors such as the competitors (e.g., as is seen in the ratio of small:large rediae for some marine trematodes; Lloyd & Poulin, 2013; Lagrue et al., 2018). It is possible that these large-pharynx rediae are an alternative response to similar selective pressures faced by freshwater and marine trematodes: small rediae (for some marine trematodes) and large-pharynx rediae (for some freshwater trematodes) may both act as ‘soldiers’ and help to defend the snail from competitors, but perhaps the ‘soldiers’ of freshwater trematodes are less specialized. Their larger body size may allow them to continue reproduction but do less to minimize pathology to the host snail (another proposed advantage of small soldiers). These traits may be more beneficial (or at least less detrimental) to trematodes infecting shorter-lived freshwater hosts (Hechinger, Wood & Kuris, 2011 suggest something similar). More data are clearly needed before any of these hypotheses can be assessed, but if it is found that these large-pharynx rediae are common in some trematode species and that they have some overlapping functions to the small rediae in trematodes of marine snails, their study may also provide context for better understanding the evolution of reproductive DOL in trematode parasites.

While much of our discussion has centered around the assumption that any differences noted between freshwater and marine trematodes derive from the differences in their ecology- especially the longevity of their host- it is worth considering whether these differences could instead be explained by patterns in the phylogeny. The marine trematodes for which DOL has been reported come from only three families (Philophthalmidae, Himasthlidae and Heterophyidae; Table 1). Poulin, Kamiya & Lagrue (2019) suggest that this represents three evolutionary origins of DOL because the two families within Echinostomatoidea that show DOL are not sister taxa (Tkach, Kudlai & Kostadinova, 2016). None of the colonies we sampled appear to belong to these three families, but there are many that fall within the same superfamilies (Echinostomatoidea and Opisthorchioidea; Table 3). It seems unlikely to us that DOL would have (potentially) evolved three independent times in marine systems and not have evolved in closely related freshwater clades (Echinostomatidae and Fasciolidae are both more closely related to Philophthalmidae than Philophthalmidae is to Himasthlidae according to Tkach, Kudlai & Kostadinova, 2016) if DOL were strongly favored in those clades. Phylogeny is certainly a confounding factor that cannot be ruled out as a possible explanation, but ecology/host longevity is at very least an equally plausible explanation.

Finally, we would be remiss not to acknowledge some of the limitations of this study. First and foremost is the very limited data for many of the COI species we collected. Freshwater trematode communities include a staggering diversity of species (e.g. Gordy & Hanington, 2019), and we made the decision to present all of the data we collected even though we recognize that our ability to draw conclusions based on only one or a few colonies of many species are limited. This was further hampered by our unfortunate failure to dissect infections in saline at the start of sampling. Nonetheless, we hope that these data provide some perspective on how commonly patterns associated with DOL in marine trematodes are seen (or not) among freshwater trematodes at large, even if conclusions about particular species are somewhat unreliable.

Second, we acknowledge that photographing live rediae did not allow for the most detailed measurements or observations. Live rediae stretch and contract, which affects the measurement of their body size (though likely not much), and their movement can make it difficult to clearly see certain body structures (e.g. appendages, pharynges, embryos) in photos. To compensate for the limited visibility of certain body structures, we used a ‘presence score’ for assessing the presence/absence of appendages and embryos, which allowed us to account for the uncertainty in our observations; also, we simply did not measure pharynges that were not clearly visible. Using fixed (possibly even stained) rediae for measurements would likely improve the precision of our results, but the lesser quality of our photos is unlikely to lead to false positives (germ balls/embryos might be missed in poor quality photos but would not appear present when they are in fact absent) or affect activity measurements, which must be performed on live trematodes.

Conclusions

We hypothesized that reproductive DOL would be uncommon in trematodes that infect freshwater snails because many freshwater snails do not live long enough to warrant investment in a nonreproductive soldier caste. Our results support this hypothesis: contrary to patterns seen in marine trematodes with apparent DOL, the small rediae in this study largely appeared to be capable of reproduction and, despite a limited sample size, did not appear to be more active or aggressive than larger rediae. Interestingly, we did document several trematode colonies from three species containing individual rediae with unusually large pharynges. This morphological variation among larger reproductive rediae may represent an alternative strategy for dealing with potential competitors.

Supplemental Information

Supplemental Information 1 Example image showing how activity measurements were taken.

Two photos of the same redia taken two seconds apart were superimposed using ImageJ’s Merge Channels function (one image red, one image green) and the largest distance moved by any part of the anterior end of the redia was measured in micrometers.

Supplemental Information 2 Representative photos of the snails collected from each family.

All snails are shown in the wells of a 12- or 24-well cell culture plate. Snail families are as follows: Top Left: Hydrobiidae; Top Right: Lymnaidae (some lymnaid snails we collected looked more like the pictured physid); Middle Left: Physidae; Middle Right: Planorbidae (some planorbid snails we collected were significantly larger than the one shown); Bottom Left: Valvatidae; Bottom Right: Viviparidae (most collected viviparid snails were too large to fit in the cell culture plate wells and looked more like the inset photo; the main photo is of a very young snail).

Supplemental Information 3 Summary of the snails collected during summer 2019 including numbers of infections.

Snails were identified to family (Snail Family) and were collected during several visits to each site (NMP, SP, TP, WR; see text for site locations) from late May through August. Reported is the total number of snails collected from each family (N Collected), the number of each snail family collected from each site (NMP, SP, TP, WR), the total number of infections from the collected snails (N Infected) and the number of infected snails whose rediae were included in the division of labor study (N Included). A few additional infections were included in the analysis from North Montpelier Pond (NMP) in October 2019 and summer 2021, but their collection data are not included in this table.

Supplemental Information 4 Detailed infection-level summary of the patterns summarized more succinctly in Table 4.

This Excel document lists the actual output from statistical tests, etc., for each colony. An explanation of column headings is available in File S5.

Supplemental Information 5 Description of column headings found in Supplemental File 4.

Supplemental Information 6 Representative photos of cercariae and rediae from each genetic group.

If the genetic group includes trematodes from multiple infected snails, all snail numbers are listed in parentheses following the group number. For groups with many representatives, the snail number(s) from which photos were obtained are bolded. Scale bars are all 100 µm.

Supplemental Information 7 Summary of redia counts, survival and reproduction in culture.

For all panels, the rediae from individual wells of a cell culture plate are shown separately. Rediae were cultured with snail cells (circles and solid lines) or without snail cells (triangles and dashed lines) in one of three media: a 1:1 mixture of Bge Medium to Medium F (red), a 3:1 mixture of Bge Medium to Medium F (yellow), or Bge Medium only (teal). We stopped counting rediae in wells without snail cells after day 33 because at that point, all rediae appeared dead. Cercaria production was tracked for the first few timepoints, but it became increasingly difficult to count cercariae over time as tissues from dead cercariae started to break down (esp. in wells without snail cells) or snail cells became increasingly dense (in wells with snail cells). (A) shows the number of large “adult” rediae over time. Fluctuations in redia counts are probably counting error, not actual changes in the redia population size because it is a closed system and small rediae did not grow into large rediae during the timeframe monitored. Wells were initially seeded with 100μL of rediae, which corresponded to about 7–22 rediae per well and accounts for the differences in redia numbers among wells. (B) shows an estimate of the survival of rediae over time, measured as the percentage of rediae that were active while being observed for 5 s. Counts of active rediae were divided by the number of redia the well started with to allow easier visual comparison among wells. (C) shows the number of small redia that appeared in some of the wells over the course of the experiment. We assume that these small redia were produced by the larger rediae in those wells.

Supplemental Information 8 Size measurements from rediae.

Metadata are in available in File S10.

Supplemental Information 9 Morphology notes and observations from the smallest and largest rediae of each infection.

A description of the data is available in File S14.

Supplemental Information 10 R code for importing redia size data and cleaning it up and metadata for File S8.

Supplemental Information 11 R code for analyzing redia volumes.

Supplemental Information 12 R code for analyzing redia pharynx sizes.

Supplemental Information 13 R code for extracting the numbers of the smallest and largest rediae from each infection.

Supplemental Information 14 R code for comparing the morphology of small vs. large rediae.

Supplemental Information 15 R code for comparing redia sizes among different body regions of the snail host.

Supplemental Information 16 R code for exploring characteristics of rediae with large pharynges and colonies containing these rediae.

We are grateful to Mary Nsubuga and Caleb Scully for helping with the genetic analysis of samples presented in this study. Additionally, we thank the students in Prof. Neal’s Fall 2022 Trematode Ecology course for their attempts to identify additional trematode colonies with substantial variation in pharynx size and Heather Driscoll of the Vermont Biomedical Research Network for her guidance with the submission of sequencing data to GenBank. The contents of this article are solely the responsibility of the authors and do not necessarily represent the official views of NIGMS or NIH.

Additional Information and Declarations

Competing Interests

Author Contributions

DNA Deposition

Data Availability

The authors declare that they have no competing interests.

Allison T Neal conceived and designed the experiments, performed the experiments, analyzed the data, prepared figures and/or tables, authored or reviewed drafts of the article, and approved the final draft.

Moira Stettner performed the experiments, authored or reviewed drafts of the article, and approved the final draft.

Renytzabelle Ortega-Cotto analyzed the data, authored or reviewed drafts of the article, and approved the final draft.

Daniel Dieringer performed the experiments, authored or reviewed drafts of the article, and approved the final draft.

Lydia C Reed performed the experiments, authored or reviewed drafts of the article, and approved the final draft.

The following information was supplied regarding the deposition of DNA sequences:

The trematode COI sequences described here are available at GenBank: OR666887–OR666945.

The following information was supplied regarding data availability:

The redia measurements, morphology notes from the smallest and largest redia of each infection and the R code used to perform the data analysis is available in the Supplemental Files.

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
