# Peer review of "Freshwater trematodes differ from marine trematodes in patterns connected with division of labor"

_PeerJ, doi:10.7717/peerj.17211_

## Round 0.1 · original submission · Minor Revisions

Thanks for considering PeerJ to submit your manuscript, and congratulations on completing this body of work. The reviewers agree that the manuscript provides important and interesting data about DOL and should be ready for publication after minor revisions. Thankfully, all reviewers provided ample constructive feedback, which should improve the current draft and eliminate any issues that might be unclear or confusing to the reader. In particular, some issues were raised regarding the quality of some of the images and, in turn, the morphology data. Additional discussion is needed regarding the switching between DI and saline and the additional context with regards to the phylogenetic differences between the taxa observed, as mentioned by the reviewers. Please address the issues raised by the reviewers in the rebuttal and the revised manuscript. I look forward to reading the revised version.

Reviewer 1 ·

Basic reporting

The writing is generally excellent, though I spotted a few typos (e.g., line 447, outliers is misspelled; lines 462-463, double negative makes the sentence confusing; etc).

The authors keep referring to an ‘anecdote’ about the potential growth of small rediae (e.g., lines 498-499). However, it has already been shown that they can grow in size, even if not ever becoming reproductive (see Kamiya et al. 2013 Evolutionary Ecology 27: 1235-1247).

Experimental design

I have reservations about the attack tests and their validity. They were conducted after all other measurements had been taken (so, not immediately after dissection), and they were made on rediae placed in tissue culture wells, in an unnatural medium (mostly in just plain water). I doubt whether the rediae were in any condition to display normal behaviour. Also, they were only allowed 30 seconds to perform attacks. I find the attack tests mostly unreliable for these reasons.

Validity of the findings

My main criticism is the fact that phylogenetic issues are not considered. Most marine trematodes in which clear division of labour has been reported belong to families (Himasthlidae, Philophthalmidae) that do not occur in freshwaters. Reproductive division of labour might be a difficult evolutionary transition, one that probably did not occur often over the phylogenetic history of trematodes. If it happened in the lineage that gave rise to certain predominantly marine families, and not in lineages of predominantly freshwater families, then the marine VERSUS freshwater comparison is confounded by phylogenetic constraints. This definitely needs to be addressed by the authors.

Additional comments

This is an interesting study testing for the occurrence of reproductive division of labour in multiple freshwater trematode species. The authors acknowledge some of the limitations of their study, and use caution to draw conclusions. Overall, the results will add to the slowly growing knowledge on this phenomenon. I do have some comments and suggestions for the authors, as indicated above.

Reviewer 2 ·

Basic reporting

All figure files, table files, and supplementary data files open and appear to be complete. GenBank entries are public and available.

Experimental design

This manuscript tests the hypothesis that freshwater trematodes have division of labor (DOL) at the redial stage in snails, similar to that described for marine trematodes. This is done by collecting snails from 6 families in freshwater habitats in Vermont, examining them for cercariae and also crushing snails to dissect them to examine for rediae. A total of 54 redial infections were used in the analysis which consisted of qualitative morphological, morphometric, and behavioral study. The authors do a very good and thorough job of describing their methods and data.

The authors are to be commended for their efforts to a thorough approach to the question. The data presented represent much planning, time, and effort. And I appreciate that the research is driven by a question and hypothesis. However, some types of data are so small in number that I’m not sure that they merit inclusion. Or if they stay included, I think they cannot remain prominent in the summary and conclusions as it cannot be said that the particular aspect was truly tested. Primarily, I am thinking of the developmental data where rediae were from a single infection. I don’t think the developmental ‘patterns’ can be mentioned in the same sentences as the morphological and behavioral because there is such a difference in species and family representation.

Validity of the findings

Though I am not familiar with the literature, it seemed that there was little reference/comparison in the discussion to the freshwater studies listed in Table 1. Another factor that I think the authors need to explicitly address is how the genera/families studied compare to those studies. For instance the one freshwater taxa in the literature that seems to have some DOL is Haplorchis pumilio, an opisthorchid. Table 3 only lists one opisthorchid, which might be the most likely place to see DOL, but it’s not clear if this taxa was heavily scrutinized in the study. Similarly, it needs to be clarified whether there is any overlap between the families of marine and freshwater trematodes. Do any of the families with marine representatives in Table 1 have freshwater species? Do any of the families with freshwater representatives in Table 1 have marine species? If there is no overlap, then taxonomy is a confounding factor in the marine/freshwater DOL dichotomy. If there are overlaps, then those families will be of high interest for examining whether DOL evolves as strategy in larger and longer-lived snail hosts.

Additional comments

Specific edits:
Line 2: make division of labor all lowercase; the justification to capitalize it in title is not apparent

Line 102: have instead of share – you can’t share differences

Line 165: It’s either Omega EZNA kit or Qiagen DNeasy kit; I don’t believe there is a Omega DNeasy kit

Line 455: outliers instead of outliners

Line 471: sentence needs work: …we do not believe that the sampled species demonstrate reproduction DOL.

Line 486: …several of the patterns…


I don’t think it is sufficient to have the GenBank numbers listed only in supplemental materials (in fact, I think they are only mentioned on the journal review site and not in any particular document?). I would also object if the range is just listed somewhere in the document – since they cover a wide range of species, individual sequences should be clearly attached to each identification. Readers should not have to hunt through the range of accession numbers searching for the specific one they want. Instead, Table 3 should contain an additional column with the GenBank accession numbers for each genetic group. This would permit anyone interested in a particular sequence based on its nearest BLAST hit and identification to find that sequenced.

Another note about Table 3 is that it should be made clear whether the 6 Unknown sequences are similar to each other or represent multiple distinct species. If the latter, they should be represented as multiple rows in the table.

·

Basic reporting

Basic reporting requirements are almost met.

1. See below for details on how additional data should be provided or aspects of data clarified.

2. I suggest more scholarship and context be provided in the introduction concerning what is known about interspecific antagonism in freshwater snails. Doing so would make even more compelling the paper under review. See line specific comments for more information.

3. The tables tend to be difficult to understand. I would take steps to make them easier to digest. One way to do this could be to spell out the entire column header (at least a full word) instead of the abbreviations. This could be done with slanted headers. Alternatively, the header term could be fully spelled out and defined in the footnotes. E.g., for Table 1’s “App.”, the footnote could say “Appendages are pronounced…” instead of “Structures are pronounced…”.

Experimental design

1. The research question is well-defined, relevant, and meaningful.

2. The research involved acquiring numerous types of data. Much data acquisition appears to adhere to a high-enough technical standard. However, there are genuine issues with several very important elements of the data. The issues are perhaps not insurmountable, and the authors are aware of them. They sometimes provide arguments as to why the problems should not be “deal breakers”, or otherwise took steps to account for them and provide transparency in the degree of certainty. I agree that the problems perhaps do not preclude publishing the data, particularly because so many aspects of the data appear to be high quality, but I would be remiss to not bring them up in the review. In other cases, I suggest that the authors provide more information or otherwise augment their argument for the issue being surmountable. Details follow in other general comments

3. There are issues concerning the quality of the worm morphology data and, particularly, the reproductive state data. The authors are aware of most of the problems, and they provided some arguments as to why the problems should be tolerable, and use an approach to clarify the degree of uncertainty for one part of the data.

What is the specific problem? Well, both worm morphological and reproductive state data were obtained from photos. That is not necessarily a problem in and of itself, but the photos appear to typically be of very poor quality—e.g., very blurry/taken out of focus, and that problem is combined with the photos being taken on live and unflattened worms. I’ll first deal with the latter two issues.

Taking photos of live rediae is a general problem for getting high quality body and pharynx size data. Live rediae often move around a lot and greatly distort their bodies (with stretching and contracting), which adds a good amount of noise to size measurements (i.e., body length and pharynx size). This is why size measurements are typically done on rediae killed in a standardized way (e.g., heat killed). However, the authors note that their rediae did not move around a lot. Perhaps that observation, and the great differences in the groups of rediae measured within each colony probably means that the noise in size measurements is dwarfed by the actual true, large differences in, say redia size. The authors may wish to add this latter point to their paper.
Being unflattened is actually good for the size measurements, but suffers for depicting “reproductive potential”. It is simply all to easy to miss small embryos (particularly germ balls) when the worms are not flattened (particularly if not heat killed and mounted in a clearing medium). And this issue is important for understanding whether the smallest rediae lacked early stage embryos. The authors made a graded scale of certainty for whether rediae had or lacked embryos, and this definitely helps. So, perhaps the ms provides useful info for ~2/3 the infections (43/61), where the authors were at least “relatively sure” that the smallest rediae had embryos. For the other 18 infections, it would seem that we simply do not know the status of the reproductive state of the smallest rediae. Table 4 presents the data in just this way, so I would argue that this data quality problem also does not preclude publishing the results.

The poor quality of the photos will also be concerning to many reviewers. This is particularly so because the example photos (Supp fig6) are quite blurry, out of focus, and low resolution photos. In fact, it is not clear to me that one can even be certain we are dealing with rediae vs sporocysts in some cases (e.g., see Unk: P139). One might think that the example photos would represent the best photos. But they are often quite poor—so what does that mean for the rest of the photos? Are they equally bad or worse? Here too, because the authors did a good job of being transparent with the degree of certainty/uncertainty. Because the “positive” findings are nearly certainly not false positives (e.g., one would not accidentally score a redia as having embryos), and because noise in the morphological data driven by worm movement probably is not influencing the most important conclusions, I would here too evaluate that this quality problem with the technical approach does not preclude publishing the paper.

Again, these problems with data quality (perhaps more clearly, uncertainty) must also be placed in the context of the other parts of the data that seem to be of high quality taken from the same individual trematode colonies (e.g., genetic sequencing of dispersive offspring, activity trials, attack trials, some morphological analyses). So, for instance, even with ambiguity about the reproductive status of the small rediae, the other data should still be valuable, further justifying publishing the paper.

4. There is a problem in that the authors used distilled water (DI) instead of saline for much of their work. The authors clarify that they were initially unaware that it is usually important to use saline to not osmotically shock the trematodes. The authors become aware of this issue, and switched to a saline solution partway through the project, and noted that doing so “substantially increased the activity levels of rediae”. The authors partly accounted for this methodological problem in the behavioral parts of their work (see below). But it appears that they did not do any assessment of whether using distilled water influenced their morphological measurements.

DI water can cause rediae to greatly swell and inflate in size. I therefore suggest that the authors explicitly address in the ms this issue concerning their morphological data, including body size, pharynx size, and the appearance of appendages (which might not be evident with a swollen body). Perhaps minimally, the possible issues should be stated and how each colony was treated should be clarified (if not already done so in the supplementary data sets). But perhaps for some of the species (particularly the one with an n of 16), the authors might be able to analyze differences between DI water and saline treated worms. It might also be possible to show that morphology wasn’t altered because photos were taken of worms shortly after dissection, before gross morphological distortion from osmotic shock would occur. But that would likely require new data and analysis.

5. The use of DI water is an even more obvious problem with the activity and attack trial methodology. The authors are aware of this flaw, and, in fact, removed 20 of the 28 activity trials from their activity analysis. But they did not remove the attack trials from the analysis. They also observed nearly no attacks. But the poor in vitro conditions probably precluded most attacks from occurring and presenting the “negative” results for all 22 infections artificially inflates those negative results. Hence, I suggest removing the attack data for the 20 colonies done in 0% saline and only providing the data for the 2 colonies done in saline. Alternatively, the authors could present the results along these lines: “In the 20 tests where rediae were osmotically shocked, we saw…”. A similar thing could be done for the activities. This form of clarity will be much less likely to lead readers to think that freshwater trematodes in Vermont simply barely move or don’t attack each other. The reality is that attack and activity rates were not adequately assessed in this study (except perhaps for 2 or 8 colonies, respectively).

6. There is a possible problem with the size-blind redia collections and whether they accurately reflect size-frequency distributions. The ms states that the authors noted no size bias with sucking rediae into pipettes, but do not explain how that was assessed. I hope they can provide more detail on why they do not think there was a bias in how readily large vs small rediae were pipetted up. Ideally there would be data. But, in the line-specific comments, I provide what might be the only “data free” type of justification. Understanding whether the presented size-frequency distributions accurately depict reality is important for understanding what information/knowledge the ms contributes.

If a good argument cannot be made that the size-frequency distributions (SFDs) at least crudely reflect reality, then they should not be depicted that way. In this case, I suggest clarifying that they DO NOT necessarily represent actual SFDs, but just what happened to be slurped up. The information on sizes found may still be useful.

7. Many readers will also find another aspect of the data disconcerting. That is, the ms reports having data for 20 species of trematode. But they only sampled a total of ~60 colonies. Most species only had a sample size of one! Therefore most species were simply not accurately characterized.

I suggest that this is also okay, but that the authors should shift the language a bit to emphasize that they studied ~60 colonies that spanned 20 species. Importantly, they should provide an argument that they are not characterizing sociality of species, but of individuals colonies. They would benefit from making the point that sociality is a characteristic of specific societies (including colonies) NOT necessarily of species. They can see, for instance, Crespi and Yanega 1995 for that point.


8. Finally, the authors morphologically identified the snail hosts, but this identification appears to have been done only to the family level. This may be okay. But, given the instrumental nature of host ID to so much of parasitology, it would be really good if the authors could provide more information on the hosts for each colony (or at least species).

Given the nature of the study, perhaps a possible, perhaps best, solution would be if they could deposit a photo of the host for each colony. If they took a photo of each host snail prior to dissection, such a supplemental file would be very helpful.

In addition, I would at least clarify for each colony what the family of the host was in the supplemental data file table 4.

Finally, if possible, it would be good to clarify if it appeared that, for those trematode species with replicated colonies, the trematodes appeared to use the same type of host snail or not.

Validity of the findings

As discussed above and below, some of the findings are fine and useful, even if some of the conclusions from the findings are made too broad. They can be adjusted. Some other findings are perhaps partly invalid as currently expressed. But additional clarity and data treatment should help here. See above and below, for the issues concerning the osmotically shocked rediae and the morphology and behavioral work.

1. Relating to the vagaries in the data discussed above, I suggest that there is too much extrapolation from the subset of the data that can be used to indicate a lack of non-reproductive soldiers. The authors conclude that none of the studied species or colonies have soldiers. But several of the species simply have no data on whether small rediae are reproductive or not. Further, there is an extremely low amount of data on activity and, particularly, attack rates. That data is instrumental for testing whether the small rediae serve as soldiers at all (even as a temporal caste). E.g., it is possible that many of the small rediae that likely have offspring may very well serve as a temporal caste. It is premature to rule that out, given the lack of viable attack trials for somewhere around 59 of the 61 colonies.

So, I suggest that the conclusions in the abstract and discussion be reworked to better recognize what can be inferred or not from the available data. The clarity achieved by doing so will make the paper more rich and more useful. See line specific comments for more.

2. I suggest a bit more scholarship to enhance the section in the discussion about the large and small pharynx reproductive rediae. As the authors note, it is entirely possible that the authors have simply encountered reproductive rediae of different generations in the colony. The first generation rediae of at least a few freshwater species are known to have larger pharynxes than subsequent generations (e.g., see Lie & Basch 1967, and the work by Rondelaud and colleagues starting in the 70s on Fasciola hepatica, in addition to the references the authors provided). We also know that sometimes rediae of the same generation can have large or small pharynxes (e.g., see Donges 1963 Naturwiss—cited in L&B67).

The other possibility (not mutually exclusive, as noted by the authors) is that the large pharynx rediae may serve as defenders. This is certainly a possibility—and is particularly interesting if they are not reproductive. If possible, I suggest that—if possible—the authors provide explicit evaluation of whether those rediae were actively reproducing or not. I realize that they are reliant on what are often sub-par photos, but they did use those such photos for other parts of the study.

Another possibility not considered by the authors is that they at least sometimes detected reproductive rediae of DIFFERENT species in the same snail. Such mixed-species colonies occur less frequently than expected, but they certainly do occur (see Kuris & Lafferty 1994 etc) and could have readily been missed in the barcoding of shed cercariae. The two rediae depicted in Figure 5D do look like they could be of two different genera, or even different families to me. I suggest the authors explicitly discuss, and evaluate if possible, this possibility in the paper.

See line specific comments for more on this topic.


3. I am not convinced of the utility in some cases of providing numbers or results pooled among all the colonies and species. I suggest sometimes providing results at the colony and species level (yes, even though species are not adequately characterized, it may still be handy to clarify, for instance, that “we were reasonably confident that the small rediae were actively reproducing in colonies of X of X species”. Another place where pooling all colonies does not seem helpful is in the total numbers of rediae in the different regions of the snails. Doing so obscures inter-colony and interspecific variation.

Additional comments

This paper deals with an interesting issue, involves a substantial amount of diverse work, the provisioning of a substantial amount of supplementary files and code, has problems with some aspects of the data, has the authors taking steps to deal with some of the problems in the data that are sometimes adequate and sometimes perhaps not (but can be), appears to have some overly broad conclusions, yet does, I think, provide useful information and an overall good introduction and discussion, both of which also could benefit from some enhancement.

In the above sections, I provide comments on possible issues with several aspects of the data—and how those issues are addressed well enough in some cases or could be addressed better in others. On the whole, I think with some minor modifications, the ms can provide useful information. Here, I provide additional comments largely focused on helping improve the ms.


1. I suggest an additional, simple analysis. If the authors can justify that the SFDs can be believed to reflect reality (see above), I suggest that they provide an analysis that compares the frequencies of being relatively certain/uncertain about reproductive “potential” for the two groups of SFDs (namely, bimodal or unimodal). It would be interesting to know if the proportion of time certainty was had (or not) concerning reproductive status was equal for colonies expressing bimodal vs unimodal SFDs. A chi-square contingency test would likely suffice here.


2. I suggest considering using a different term than “genetic group” to refer to the probable species (defined on DNA sequence similarity). Why not say “species” (perhaps w/ the quotes), or “CO1 species”, or “phylogenetic species”? Any of those terms seems to align more clearly with what we think the groups represent. One might think ‘genetic group’ referred to members of clone, for instance, not members of a species.




Line specific comments:

23 From abs: “Here, we present data from twenty species of freshwater trematode
24 exploring nine morphological, behavioral, and developmental patterns…”
See general comments for rewording this to clarify that ~61 colonies were studied (that involved 20 species) to not mislead reviewers into thinking that biological attributes of 20 species were adequately depicted (given the lack of replication within species).
Also, I would suggest rewording to accurately reflect that the nine patterns were not studied for most of the colonies (e.g., attack trials only adequately done for 2 colonies). See general comments.



43 from abs “We conclude that these freshwater species likely do not have a group of
43 specialized non-reproductive soldiers because small rediae do produce embryos and do not show obvious behavioral differences from larger rediae.”

See general comments: I am not seeing how one could fully adopt this conclusion given the (1) uncertainty in whether small rediae were reproductively active or not for at least four of the sampled phylogenetic groups and because the behavioral differences were simply not adequately assessed for the bulk of the studied colonies. Some sort of qualification should be done to clarify the subsets of the colonies for which firm conclusions can be made.



65 “Clonal proliferation in the snail produces a population (often large)…”
I suggest attaching some numbers to “large” as large will mean very different things to different people.


71 “Evidence suggests that the community of trematode species that
72 utilize the same host snail species interact with one another, even forming complex dominance 73 hierarchies (e.g. Kuris 1990, Fernandez and Esch 1991, Sousa 1993). 73 Redia-producing 74 trematodes often have a competitive advantage in these hierarchies (Sousa 1992), and 75 producing a specialized soldier class may grant an additional benefit, allowing these species to 76 reach a dominant position in the hierarchy (Metz 2022).”
…”

I recommend checking and citing (if the authors agree that it makes sense) Kuris and Lafferty (1994, ARES). It is perhaps the single best citation to include here—because they generated dominance hierarchies for numerous trematode guilds, conducted the most extensive analysis (62 published datasets) of the impact of interspecific competition, used arguably the best analytical approach, and because they provide perhaps the most extensive review of “larval” trematode antagonism (including lit throughout the 1900s) that may provide the best entry for people wishing to go further. That prior literature includes the foundational work on dominance hierarchies by Lie and colleagues from the 1960s and 70s. [I think it’s okay to not directly cite that foundational work in lieu of citing Sousa 92 and K&L 94.]

In the second sentence I recommend considering adding Kuris 1990 along w/ the cited Sousa ‘92—as I think that chapter provided a more detailed review of the experimental and review literature of Lie and colleagues and explicitly listed the point about redia being dominant to sporocysts.




73 “Redia-producing 74 trematodes often have a competitive advantage in these hierarchies (Sousa 1992), and producing a specialized soldier class may grant an additional benefit, allowing these species to 76 reach a dominant position in the hierarchy (Metz 2022)…”

Hmmn. I do not think that it is a good idea to cite the Metz 2022 thesis here. That thesis was done by a student of mine. Not only is it still currently embargoed, but we do not make any point (I’m pretty sure) about soldiering helping species “reach” a dominant position in a dominance hierarchy. In one of his chapters, we do document numerical ecological dominance by a species that also holds a dominant position in the dominance hierarchy (and which has soldiers). We do make the basic point that its ability to defend (with soldiers) helps it reach that ecological dominance. But ecological dominance (e.g., being the numerically most common member of a guild) is different than holding a dominant position in a dominance hierarchy.

If the authors wish to connect dominance in a dominance hierarchy to soldiers here, something like the following should work: “Redia-producing trematodes often have a competitive advantage in these hierarchies (Sousa 1992), and competitively dominant species may tend to evolve soldiers (Hechinger et al 2010).”

Hechinger et al 2010 do broach the above hypothesis.




78 “ Hechinger et al. (2011; and shortly thereafter Leung and Poulin, 2011) were the first to propose 79 trematodes had division of labor when they demonstrated two distinct size classes of rediae and 80 highlighted several differences between them…”

In the remainder of the paragraph started with this sentence, I would add in the single, most important evidence presented for soldiers being soldiers—the much greater attack rates by soldiers compared to reproductives quantified in the in vitro attack trials.

By the way, those data combine with the noted observations of seeing soldiers aggregating around and attacking invaders in situ (in freshly dissected snail tissue-see also Garcia et al 2017 for that ‘observation’) and the broader ecological context of substantial interspecific competition (e.g., the Sousa and Kuris/Lafferty papers cited above) in the system to make it nearly impossible for those soldiers to not be soldiers (temporary or permanent). Garcia et al 2017 perhaps most clearly lay out that missing “ecological context” argument.




82 “Small rediae are active, slim, and free of developing embryos, but possess large (relative to their bodies) muscular pharynges in their 84 mouthparts. The authors proposed that these small rediae forgo reproduction (at least for a 85 time; see Metz 2022) to act as soldiers”

In addition to adding a sentence clarifying that soldiers actually attack at much greater rates than reproductives, I don’t see a good reason to cite the embargoed Metz thesis here. I suggest a rewrite along these lines:

“Small rediae are active, slim, and free of developing embryos, but possess large (relative to their bodies) muscular pharynges that they use to attack at least heterospecifics at much greater rates than the large rediae. The authors proposed that these small rediae forgo reproduction to act as soldiers (although the temporal or permanent nature of the caste remains unresolved (Mouritsen and Halvorsen 2015 Mar Bio; Garcia et al 2017 IJP)).”






96 Freshwater trematodes share both key similarities and key differences with marine/estuarine
97 trematodes. One important similarity is that the intramolluscan stages of all trematodes live in
98 clonal groups that share a localized, defendable resource (the snail), which favors the evolution 99 of DOL (Hechinger et al. 2011). Also, both freshwater and marine trematodes face invasion of 100 the snail by other trematodes, and once invaded, competition is likely intense (Sousa 1992). 101 The ability to directly attack one.s competitor provides an advantage that could be increased by 102 a specialized class of soldier rediae (Poulin et al. 2019).

Here, I suggest that it is important to note that interspecific antagonism is well-known to occur among freshwater trematodes. The original work by Lie (e.g., Lie 1965 nature) and colleagues (see refs in Sousa and Kuris/Lafferty papers above) were done on freshwater trematodes. Further, Lie (1969) described “immature rediae” that defended a snail (cited in Hechinger et al 2010).

So, the stage is set for possible interesting and diverse findings to be found in freshwater trematodes. Noting this context makes this paper under review even more compelling.






104 One important difference between marine and freshwater trematodes is the average longevity of 105 their host snails. Freshwater snails tend to be short-lived in comparison with marine snails 106 (Heller 1990), and a short-lived host reduces the benefit of defending the snail host from 107 potential competitors (Hechinger et al. 2011, Poulin et al. 2019): if the host will die soon anyway, 108 there may be no benefit of diverting resources to a specialized soldier class; it is likely more 109 prudent to instead invest in quick production of dispersal stages.


I would probably add in that Hechinger et al 2010 also noted that shorter longevity could select for larger totipotent rediae that defend and reproduce (instead of small soldiers). That is, short life span may not only influence being able to defend or not, but also the type of defense. The idea of large totipotent rediae clearly bears on findings in this ms!


Also, I would change “no benefit” to “less benefit”. Or make it “no net benefit”.










149 Please clarify the type of lettuce used for food.




187 “This was performed without observing what size rediae were being collected,
188 but when individual rediae were collected with a pipette it did not seem that any size redia were 189 more difficult to collect, so this method should be relatively unbiased.”

Hmmn. I think more information on how this was assessed is important, given that there frequently are differences in how readily larger/heavier and smaller/lighter/sometimes stickier rediae are sucked up by a pipette. Ideally, of course, one would have data showing the lack of bias. But, a realistic alternative justification could be something like: “For each pipet draw, we took ALL rediae in a certain area of the dish”. This would also reflect a lack of bias.





206 “and 207 cercaria/embryos/germinal balls (to assess reproductive potential).”

I would clarify the approach here. Are you following the approach of Hechinger et al 2010, which scored number of embryos free in a brood chamber? This is important to clarify because there could be very early stage germ balls/embryos, say, in a germinal mass embedded in parenchyma tissue at the posterior of the worm (even before the brood chamber has started to open up). I consider the “free germ ball/embryo” criteria to be a fine metric, but the operational criteria should be stated.








226 Attack Tests
227 Again for colonies with sufficient rediae, we assessed the tendency of smaller vs. larger rediae228 to attack colony mates, conspecifics and heterospecific trematodes. For this test, we selected229 the 20-30 largest and 20-30 smallest redia remaining after the previous two tests and split them230 into three cell culture plate wells per redia size.

Please explicitly provide the time lapse (at least approximate) between dissection and running these attack trials, which is important give the general tendency for rediae to become less active with increasing time post dissection.






247 In the “Culture” section, I would indicate how the techniques follow or are modified from prior in vitro culture techniques. Although the “new” Lloyd and Poulin (2011) marine trematode medium is referenced, I do not see refs for the prior developed FW snail/trematode media/techniques.




291 Correct “Wilcox” to “Wilcoxon” here and everywhere else.




321-332 Most of all of this first results pgraph seems like it would likely be best off incorporated into the methods.



334-340 I suggest also putting this second results paragraph into the methods.



349: If the authors can provide a justification for the size-frequency distributions accurately reflecting reality (see above), then I suggest renaming this subsection “size-frequency distributions” instead of “volume”. If they cannot provide a reasonable justification for the size-frequency distributions representing reality, then I’d change it to “body volume”.



383 “Combining all infections, a total of 2,277 small and large rediae were isolated from snails. Gonad 384 region, 778 from the middle region, and 50 from the foot region. (Note: another 1031 small and 385 large rediae did not have their body region of origin identified and 3110 rediae were excluded 386 because they belonged to excluded infections [e.g. due to labeling issues, see above] or were 387 not classified as either small or large). In this combined sample, only 26% of rediae with 388 assigned size in the gonad were small.”

Because distributional pattern is a colony-level and likely a species-level character, it is very unclear why it would be useful to present the patterns for all data pooled. That would blur any real differences among colonies or species. I recommend shifting focus to colonies and species.








461 “Based on our preliminary analysis of trematode infections representing twenty or more species
462 (our genetic groups likely correspond to species) from six or more superfamilies, we do not
463 believe that the sampled species likely do not demonstrate reproductive division of labor (DOL).”

See general comments: I suggest that this conclusion involves too much extrapolation from the subset of species that have enough data to support the conclusion. It should be modified to more precisely represent what can be inferred (e.g., no non-reproductive soldiers, and no temporal caste soldiers, vs simply unknown.)




555 “It is possible that these large-pharynx rediae are an
546 alternative response to similar selective pressures faced by freshwater and marine trematodes: 547 small rediae (for some marine trematodes) and large-pharynx rediae (for some freshwater 548 trematodes) may both act as .soldiers. and help to defend the snail from competitors, but 549 perhaps the .soldiers. of freshwater trematodes are less specialized. Their larger body size may 550 allow them to continue reproduction but do less to minimize pathology to the host snail (another 551 proposed advantage of small soldiers).”

I think the authors are expressing a good hypothesis here. As noted above, this line of reasoning, and being familiar with other reports in the literature of large and small pharynx rediae (e.g., see Lie & Basch 1967; Donges 1963), is partly what drove me to write the line about totipotent (defensive/reproductive) rediae possibly being selected in shorter lived snails in Hechinger et al 2010. But, in that paper, we did not explicitly tease things out like the authors are doing here so it is well worth doing!

---

## Round 0.2 · Minor Revisions

Thanks for resubmitting the manuscript. The reviewers agree that the manuscript is almost ready for publication. A few minor issues appeared in the revision that need to be addressed prior to publication, in particular the 4 points mentioned by Reviewer number 3. I look forward to reading a revised version of the manuscript and a response to the minor issues remaining.

Reviewer 1 ·

Basic reporting

See my general comments below.

Experimental design

See my general comments below.

Validity of the findings

See my general comments below.

Additional comments

The authors have satisfactorily addressed the issues I raised in the earlier version. I have no further comments, and I now recommend acceptance.

Reviewer 2 ·

Basic reporting

The article is improved through suggestions from reviewers and good effort by the authors.

Experimental design

The authors note limitations to their studies and have clarified and in some cases increased their cautionary statements where appropriate.

Validity of the findings

No comment

Additional comments

Line 69 – temporary instead of temporal. Temporal can mean both “relating to time (as opposed to space)”, or as “lasting only for a time”. Temporary, however, is less ambiguous, and only has the one meaning, “lasting only for a time”. Journal editors can weigh in here.

Line 136 – same

Line 965 – remove one of the ‘the’s

Line 1006 – remove ‘we would argue that’

·

Basic reporting

Near pass, minor clarifications and discussion point should be added (see other comments).

Experimental design

Pass (with extra clarification of an issue with attack trials).

Validity of the findings

Pass

Additional comments

It appears to me that the revision adequately responds to most issues brought up in the previous reviews. Such responses include particularly the shift to strongly emphasize the reliance of conclusions on the detection of reproduction in small (young) rediae and in being clearer about the limitations or possible issues concerning the activity and attack data. The revision also appears to have adequately addressed more minor issues: e.g., (1) discussion of the improbable nature of phylogenetic confounds driving results, (2) diminishing the importance of the limited data from culture, and (3) clarifying the table and other misc. information.

I now have four things to bring up, all involving issues brought up in the original reviews or which come up given the responses. They involve easy modifications that I suggest be implemented. One stems from a problem that came to light in one of the author responses to the reviews (and requires further diminishment of the attack trial data). The other involves adding perhaps a sentence or two to help avoid having readers dismiss out of hand the data stemming from the photos. The third has to do with responding better to Reviewer 2’s point about a seeming lack of comparison of the current findings to those concerning other species. The fourth issue is related to the third, and has to do with whether it is appropriate to include the information stemming from the embargoed dissertation. As I said, I think the fixes are relatively easy to implement, and will result in improving the ms.


All of the attack trials unfortunately are likely unreliable.

While clarifying for Ref 1 that they had not provided only 30s time for rediae to interact before scoring attacks, the authors clarified that they had provided “several minutes” for rediae to interact and attack before scoring attacks. However, several minutes is still nearly certainly not enough time to consistently observe attacks. It is all too likely that one would observe zero attacks in many replicates where attacks would have occurred with more time for these slow animals.

Reflecting the time perhaps typically required to permit a reasonable chance of seeing attacks, all prior work quantifying attack rates for soldier rediae provided 90-120 minutes time (see Hechinger et al 2010, Miura 2012, Nielsen et al 2014, and Garcia-Vedrenne et al. 2016 & 2017).

This problem of this short interaction time combines with the previously identified issue that most of the rediae were osmotically stressed. That osmotic stress could render unreliable the attack trials for 20 of the 22 infections, even with ample interaction time. Now, with the lack of interaction time clarified, we must think that even the trials involving the two infections that used saline are unreliable.

However, I still think there is value in providing the attack trial info. Attacks by rediae were observed in one of the two trials using saline, despite there only being several minutes provided. Observed attacks occurred for both large and small rediae in that one case. There were also sporadic attacks by large and small rediae in the osmotically stressed samples, Hence, there is a HINT that the small rediae do not disproportionately attack (as they do for species with soldiers).

So, to include the attack trials in the paper, I would suggest that the reliability issues should be upfront, crystal clear, and difficult for readers to miss. The revision has improved on this front—e.g., it now clarifies the small sample size limitation in the abstract’s results. I suggest that the probable insufficiency of the attack trials also be included there. Perhaps it would suffice to enhance their parenthetical “(admittedly based on a small sample size)” as such: “(admittedly based on a small sample size and possibly insufficient attack trial methodology”).

This would seem to be a pretty easy pill to swallow given the shift in emphasis to the probable reproductive development observed in small redia as being the substantial findings of the paper.




I do not think that they have adequately responded to my large effort grappling with why the poor quality of the photos should not preclude publishing data based on those photos. But this is an easy fix.

The lack of adequate response is partly my fault, as I did not end that section with a “I suggest that they should do this...” I suggest that being transparent about the degree of uncertainty (as the authors are) is not enough. I suspect that some readers would not go through the effort to find the utility in the data—they will dismiss the data out of hand. I therefore suggest that the authors add a succinct bit laying out how the poor-quality photos would not have driven false positives or caused problems with the activity analyses. This is important because the most important conclusions stem from the photos. A good spot seems to be the last discussion pgraph before the conclusions, which already discusses the problems with the photos.



Possibly including a little more discussion relating current findings to prior findings involving freshwater trematodes.

Reviewer 2 reasonably called for more discussion relating current findings to what has what is known for the other freshwater species listed in Table 1. The authors stated that they were unsure what more they should say for two reasons. First, two of the prior studies had limitations. They stated, for instance, that “Garcia-Vedrenne et al. 2016 did not compare characteristics of larger and smaller rediae”.

However, I suggest that it would be useful for the authors to add in some comparison to those prior findings. For instance, Garcia-Vedrenne et al 2016 did assess the size frequency distributions, found no bimodality but unimodality, and also assessed the rare small immature rediae and found them to be actively ramping up reproduction. Hence, there are differences perhaps in the SFD and similarities in not detecting small non-reproducing rediae with the current paper’s findings that would be well discussed. The authors might then point out that they can not compare relative pharynx size or attack rates, as Garcia-Vedrenne et al did not single out those small maturing reproductives for such comparisons.

The second reason the authors were unsure of what more they could say is that the findings of one of the listed species comes from an embargoed phd thesis. This takes me to the fourth issue below.



Whether findings from the embargoed dissertation concerning Haplorchis pumilio (“hapu”) should be included.

I have considered a range of options, but think that all reference to the findings concerning hapu be removed from the ms. The findings come from a cited dissertation that is embargoed and the findings have not yet undergone peer-review. I think it was a well-intentioned mistake to include them, and a well-intentioned mistake for permission to have been given. But the manuscript cannot appropriately or reasonably integrate the findings of that work. Nor can readers consult the work, which is one of the basic reasons for having citations. So, it is minimally awkward and arguably inappropriate to include the info on hapu.

It should be quite easy to remove those findings, as they are not instrumental to the current paper. In fact, removing them even makes the current ms more compelling, given the dearth of info on freshwater trematodes and division of labor, given the removal of the strange partial inclusion of the information (noted by Ref 2).

I reiterate that there appear to have been only good intentions in including the information. I simply think that it was a mistake, and that it would be most appropriate and better for the current ms to not include it.

Given the unusual circumstances, I am happy to discuss the issue concerning the embargoed findings with you and the authors. I should mention two things. First, the ms including the relevant embargoed findings is currently under review (my former student and I are the authors). Second, I have consulted with my former graduate student, the person who initially gave permission to include that information from his embargoed thesis, and he agrees with me that it would be best to remove that information.

---

## Round 0.3 · accepted · Accept

Thanks for making the requested changes, the manuscript is now ready for publication.